# Towards a Unified Framework
# for Reference Retrieval and Related Work Generation

**Zhengliang Shi**[1]  **Shen Gao**[1*]  **Zhen Zhang**[1]  **Xiuying Chen**[3]
**Zhumin Chen**[1]  **Pengjie Ren**[1]  **Zhaochun Ren**[2*]

[1]Shandong University, Qingdao, China  [2]Leiden University, Leiden, The Netherlands
[3]Computational Bioscience Reseach Center, KAUST

shizhl@mail.sdu.edu.cn  shengao@sdu.edu.cn
xiuying.chen@kaust.edu.sa  zhen.zhang.sdu@gmail.com
chenzhumin@sdu.edu.cn  jay.ren@outlook.com  z.ren@liacs.leidenuniv.nl

## Abstract

The task of related work generation aims to generate a comprehensive survey of related research topics automatically, saving time and effort for authors. Existing methods simplify this task by using human-annotated references in a large-scale scientific corpus as information sources, which is time- and cost-intensive. To this end, we propose a Unified Reference Retrieval and Related Work Generation Model (UR$^3$WG), which combines reference retrieval and related work generation processes in a unified framework based on the large language model (LLM). Specifically, UR$^3$WG first leverages the world knowledge of LLM to extend the abstract and generate the query for the subsequent retrieval stage. Then a lexicon-enhanced dense retrieval is proposed to search relevant references, where an importance-aware representation of the lexicon is introduced. We also propose multi-granularity contrastive learning to optimize our retriever. Since this task is not simply summarizing the main points in references, it should analyze the complex relationships and present them logically. We propose an instruction-tuning method to guide LLM to generate related work. Extensive experiments on two wide-applied datasets demonstrate that our UR$^3$WG outperforms the state-of-the-art baselines in both generation and retrieval metrics.

## 1 Introduction

The automatic *related work generation* (RWG) system, which aims at generating a related work section for a target paper, may help the readers go through the cutting-edge research progress (Hoang and Kan, 2010). Two main adjacent stages are considered fundamental disciplines in RWG task: (1) selecting related reference papers and (2) figuring out the logical relation to write a summary to present the evolving process of a specific field.

---

\* Corresponding author.

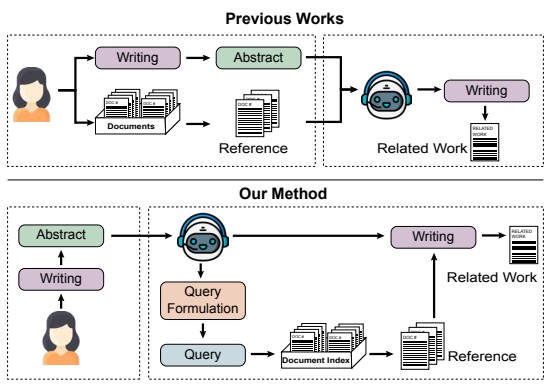

Figure 1: Comparison between the existing related work generation methods and our UR$^3$WG. We propose to retrieve references and generate related work without human-annotated references in a unified model.

However, most existing RWG methods (Chen et al., 2022a, 2021; Wang et al., 2019) only focus on the second stage, where the human-annotated reference papers are taken as input via a summarization model. Consequently, retrieving related papers from the large-scale corpus remains underemphasized. To address this problem, we focus on amalgamating the above two stages: retrieving the related papers and generating the related work. Our unified model, unlike previous RWG research, alleviates the user's burden by requiring only a high-level abstract of the target paper as input. This approach minimizes the workload for practical purposes.

An intuitive retrieval practice in RWG is taking the abstract as a query and employing dense retrieval methods to obtain the related papers according to semantic similarity. However, the abstract given by the user is too vague to be a clear intention for reference. Some related topics are inherently correlated with the paper but are not explicitly mentioned in the abstract, indicating semantic mismatches. Moreover, the presence of rare proper nouns, *e.g.,* specialized terminologies in scientific papers, makes it difficult for existing dense

retrieval methods to capture the semantic nuances. To generate the related work section, a common-used alternative is leveraging the summarization models (Chen et al., 2023a,b). Nevertheless, an ideal related work section should reason intricate relations among various references, instead of simply enumerating their main contributions. Therefore, training the model to grasp the concept of RWG while producing a high-quality related work section remains challenging.

In this work, we leverage the large language model (LLM) to enhance retrieval and generation stages in the RWG task to tackle the above challenges. As shown in Figure 2, we propose an **Unified Reference Retrieval and Related Work Generation framework (UR$^3$WG)**. Specifically, we leverage the LLM to extend the vague abstract and generate a query containing more relevant background information since the LLM has shown strong knowledge association ability (Wang et al., 2023). The generated query is used to subsequent retrieval stage. To mitigate the diminished performance of dense semantic retrieval methods, we propose a **Lexicon-Enhanced dense Retrieval (LER)** method to integrate the advantages of lexical retrieval, where a learnable lexical text-matching algorithm is introduced. We also propose a multi-granularity contrastive learning method to optimize our retriever, including group-wise and pair-wise contrast. Since the related work section should comprehensively introduce the relationship between references, instead of simply summarizing the main points, we propose an instruction-tuning method to guide LLM to generate a high-quality related work section. Extensive experiments conducted on two benchmark datasets show that our model significantly outperforms all the strong baselines, *e.g.,* pushing the ROUGE-1 to 31.59 (8.4% relative improvement) and BERTScore to 0.70 (6.9% relative improvement).

Our contributions are as follows: (1) We propose the UR$^3$WG, the first unified reference retrieval and related work generation model in the RWG task. (2) We propose a lexicon-enhanced dense retrieval method with an LLM-based query extension method to retrieve highly related reference papers, supervised by multi-granularity contrastive learning. (3) We propose an instruction-tuning method to incorporate the references and guide the LLM to generate related work logically. (4) Experimental results on two benchmark datasets show the superi-ority of our proposed model.

## 2 Related work

### 2.1 Related work generation

Similar to multi-document summarization, the task of related work generation (RWG) which usually takes multiple reference papers as input and generate a related work section for a target paper (Chen et al., 2022a; Hoang and Kan, 2010) via summarizes the related information in a logical order. Existing RWG methods can be divided into extractive and abstractive methods. Specifically, extractive methods select a subset of words or sentences most relevant to the input abstract to form the final related work (Hoang and Kan, 2010; Hu and Wan, 2014; Deng et al., 2021) . With the emergence of neural-based models (Lewis et al., 2019; Raffel et al., 2019; Touvron et al., 2023), more abstractive methods are utilized to solve the RWG task (Zhao et al., 2020a; Chen et al., 2022b). For example, Chen et al. (2022a) leverage the information of the target paper and propose a target-aware graph encoder to model the relationship between reference papers. Although these methods have shown exemplary performance, they rely on the human-labeled references in the target paper, which still requires a lot of manual retrieval work.

### 2.2 Retrieval argument generation

Information retrieval has been widely used in many knowledge-intensive natural language generation tasks, *e.g.,* question answering (Guu et al., 2020; Gao et al., 2023) and knowledge-grounded dialogues (Zhao et al., 2020b; Meng et al., 2020; Li et al., 2021). For example, Lewis et al. (2020) combine parametric and non-parametric knowledge, where the former employs the pre-train language model as a knowledge base while the latter is a dense vector index of Wikipedia, accessed with a pre-trained neural retriever. Izacard and Grave (2021) propose to first retrieve passages via sparse, *e.g.,* BM25 or dense retrieval, *e.g.,* DPR (Karpukhin et al., 2020) to obtain the relevant passages. Then a sequence-to-sequence (seq2seq) model is employed to generate the answer based on the concatenation of the representations of all the retrieved passages.

These retrieval-augmented methods are usually designed for scenarios with an explicit and clean retrieval goal. However, there have been few attempts at the RWG task. On the one hand, the model only

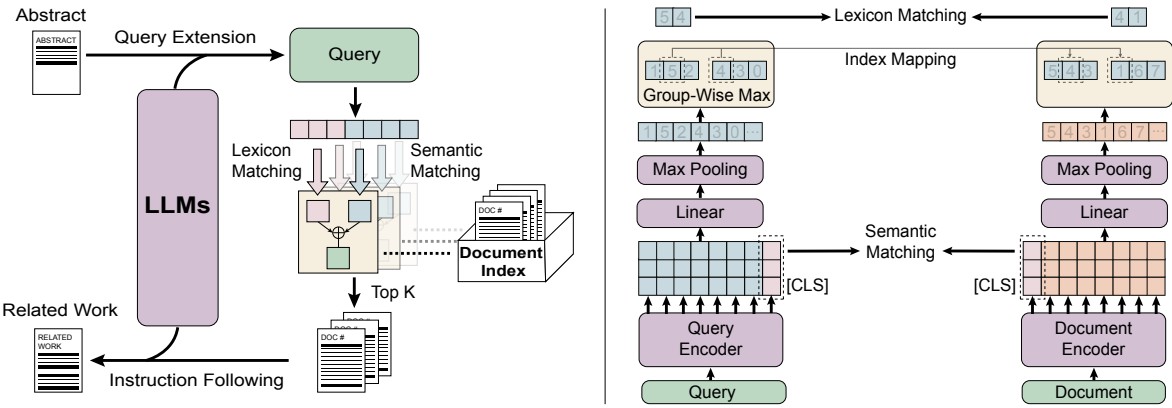

Figure 2: **Left**: The architecture of proposed framework for reference retrieval and related work generation. **Right**: the **L**exicon-**E**nhance Dense **R**etrieval (LER).

uses the abstract as input, and the abstract does not indicate which topic should be surveyed in the related work. On the other hand, it is noteworthy that the citation relation among multiple documents is implicit in nature, thereby necessitating the reasoning to understand.

## 3 Task formulation

In this paper, we generate the related work section of the target paper and only use the abstract $X$ as the input without the ground-truth references $\mathcal{R}^*$. Specifically, we propose a unified reference retrieval and text generation framework, which consists of three steps: (1) leverage the knowledge of LLMs to extend the input abstract and generate the query used for the subsequent retrieval stage. (2) retrieve reference papers $\mathcal{R} = \{r_1, r_2, ..., r_{|\mathcal{R}|}\}$ from the document corpus $\mathcal{D} = \{d_1, d_2, ..., d_{|\mathcal{D}|}\}$ according to the similarity $S$ to query at semantic level $S_s$ and lexicon level $S_l$; (3) generate the related work section $\hat{Y}$ of the target paper by summarizing the retrieved references $\mathcal{R}$.

## 4 Methodology

### 4.1 Model overview

As shown in Figure 2, UR³WG first generates the reference retrieval query by extending the abstract $X$. Then a *lexicon-enhanced dense retriever* (LER) retrieves relevant references by using the extended abstract as the query. Since RWG is different from simple text summarization tasks, the logical relation between input abstract and references should be carefully considered. We employ an instruction-tuning method to train LLM to understand the def-

inition of the RWG task and generate the high-quality related work section.

### 4.2 Query extension

It has been proven that achieving better retrieval performance is feasible if the query can be rewritten to a more similar form to the candidate documents (Yu et al., 2022). Since the LLMs trained on massive textual corpora have shown strong knowledge association ability (Wang et al., 2023), we leverage the knowledge of LLM to extend the abstract $X$ and formulate pseudo references $\hat{\mathcal{R}}$ as an extended abstract, which contains more rich information in the RWG task. The $\hat{\mathcal{R}}$ is taken as a query to search relevant references accurately. Specifically, we first design an instruction $\mathcal{I}_{QE}$ which contains two parts: (1) $\mathcal{D}_{QE}$ is a demonstration example which describes the definition of the query extension task, and (2) $X$ is the abstract of the target paper:

$$\mathcal{I}_{QE} = \text{Demonstration} : \{\mathcal{D}_{QE}\}$$
$$\text{Input} : \{X\}.$$

Then, we leverage the LLM to generate pseudo references $\hat{\mathcal{R}}$ on the condition of instruction $\mathcal{I}_{QE}$. Since $\hat{\mathcal{R}}$ is highly relevant to the abstract $X$ and can be potentially cited, we take $\hat{\mathcal{R}}$ as the query during the subsequent retrieval stage, assisting in query disambiguation and guiding the retriever (Wang et al., 2023). And we optimize the proposed UR³WG via standard language modeling objective:

$$\mathcal{L}_{QE} = \sum_{i=1}^{|\mathcal{R}^*|} \sum_{t=1}^{|r_i|} \log P_\varphi(r_i^{(t)} | r_i^{(<t)}, \mathcal{I}_{QE}), \quad (1)$$

where the $\varphi$ is the learnable parameters and $\mathcal{R}^*$ is the set of ground-truth references.

## 4.3 Lexicon-enhanced dense retrieval

Current trainable dense retrieval models built on language models have shown remarkable capability in capturing sentence-level similarity. Despite these advances, they exhibit diminished performance when retrieving references due to their omission of significant local phrases and entities (Zhang et al., 2023), which leads to outcomes worse than those obtained via traditional lexicon-based retrieval methods (Shen et al., 2022; Thakur et al., 2021; Lin and Ma, 2021a). To alleviate this problem, we introduce a **Lexicon-Enhanced dense Retrieval (LER)** method that explicitly incorporates importance-aware lexicon representations into dense semantic representations. We also design an approximation to accelerate this method for application in the large-scale corpus.

### 4.3.1 Semantic-oriented retrieval

Inspired by the dense retrieval methods (Khattab and Zaharia, 2020; Santhanam et al., 2023), we employ dual independent dense encoders: $Encoder_q$ and $Encoder_d$, which map the query $q$ and candidate reference $d$ (a.k.a., document) to token-level representation, respectively. Specifically, we add the [CLS] at the beginning of $q$ and $d$. The query representation $\mathbf{H}_q$ is obtained via:

$$\mathbf{H}_q = \text{Encoder}_q(q), \qquad (2)$$

where $\mathbf{H}_q = \{h_q^1, h_q^2, ...h_q^{|q|}\} \in \mathbb{R}^{|q| \times m}$. The $h_q^i$ indicates the representation of each token, and $m$ is the feature size. Similarly, we derive the token-level representation of each document $d$ through:

$$\mathbf{H}_d = \text{Encoder}_d(\text{d}), \qquad (3)$$

where $\mathbf{H}_d = \{h_d^1, h_d^2, ...h_d^{|d|}\} \in \mathbb{R}^{|d| \times m}$, and $|d|$ is the length of the document. We take the embedding of [CLS] token as the semantic representation. The semantic similarity between query $q$ and document $d$ can be defined based on the inner product:

$$S_s(q, d) = \text{Linear}(h_q^1)^T \cdot \text{Linear}(h_d^1), \quad (4)$$

where Linear denotes a linear layer with an activation function to compress and extract features from the raw representation.

### 4.3.2 Lexicon-oriented retrieval

To capture the information of local phrases and entities, we propose a token-level interaction network with a guided interaction mechanism that calculates the lexicon-level similarity. The architecture is illustrated in Figure 2. Specifically, we construct a continuous bag-of-words representation for each token in $q$ based on $\mathbf{H}_q$:

$$\mathbf{E}_q = \text{softmax}(\mathbf{H}_q \mathbf{W}_q) \in \mathbb{R}^{|q| \times |\mathcal{V}|}, \qquad (5)$$

where $\mathbf{W}_q \in \mathbb{R}^{m \times |\mathcal{V}|}$ is the trainable parameter, and the $|\mathcal{V}|$ is the vocabulary size. The $\mathbf{E}_q$ indicates the importance of the input token to all the vocabulary. Then a lexicon representation of query $q$ can be constructed via a max-pooling layer along each token of $\mathbf{E}_q$:

$$\theta_q = \text{MaxPooling}(\mathbf{E}_q) \in \mathbb{R}^{|\mathcal{V}|}. \qquad (6)$$

Similarly, we obtain the lexicon representation $\theta_d$ for the document $d$ via:

$$\mathbf{E}_d = \text{softmax}(\mathbf{H}_d \mathbf{W}_d) \in \mathbb{R}^{|d| \times |\mathcal{V}|}, \qquad (7)$$

$$\theta_d = \text{MaxPooling}(\mathbf{E}_d) \in \mathbb{R}^{|\mathcal{V}|}, \qquad (8)$$

where the $\mathbf{W}_d \in \mathbb{R}^{m \times |\mathcal{V}|}$ is the trainable parameter.

Although the bag-of-words representation contains fine-grained importance of lexicon, the representation $\theta_q$ over the entire vocabulary is too sparse and contains much noisy information. Therefore, we propose a **group-wise local threshold mechanism** that partitions the sparse representation $\theta_q$ into several groups containing $k$ tokens, and only remains the maximum value of each group:

$$\widetilde{\theta_q} = \{\max(\theta_q^{i \sim i+k}) | i = nk, n \leq \frac{|\mathcal{V}|}{k}\}, \qquad (9)$$

$$\tau_q = \{\text{argmax}(\theta_q^{i \sim i+k}) | i = nk, n \leq \frac{|\mathcal{V}|}{k}\}, \quad (10)$$

where the $\tau_q$ is the index of the maximum of each group, indicating the most important lexicon for query $q$ in this group. And the interaction between the sparse representation of $q$ and $d$ can be guided by $\tau_q$ to calculate the lexicon-level similarity:

$$S_l(q, d) = \sum_{i=0}^{|\tau_q|} \widetilde{\theta}_q^i \times \theta_d^{\tau_q^i}. \qquad (11)$$

Finally, we combine the similarity score of semantic and lexicon levels as the final matching score for the query $q$ and document $d$:

$$S(q, d) = S_s(q, d) + S_l(q, d). \qquad (12)$$

For each query $q$, we retrieve the top-$k$ candidate reference $\mathcal{R}$ as input. Considering the scientific paper corpus is large-scale in real-world scenarios while the online interaction mentioned in Equation 11 is time-intensive, we design an approximate algorithm for offline matching. The details and qualitative analysis are provided in Appendix A.1.

### 4.3.3 Multi-granularity contrastive learning

Unlike the existing passage retrieval task which assumes there is only one relevant document for a query (Karpukhin et al., 2020; Khattab and Zaharia, 2020), in the RWG task, a scientific paper has multiple diverse references. We propose a multi-granularity contrastive learning method to alleviate this problem, including group-wise and pair-wise contrast. Specifically, we denote the reference cited in the target paper as $\mathcal{R}^+ = \{r_1^+, r_2^+, \ldots, r_{|\mathcal{R}|}^+\}$ and the irrelevant paper as $\mathcal{R}^- = \{r_1^-, r_2^-, \ldots, r_{|\mathcal{R}|}^-\}$.

**Group-wise contrastive learning.** Given the abstract $X$ of the target paper, the goal of our group-wise contrastive learning is to minimize the distance between the abstract $X$ and the positive reference group $\mathcal{R}^+$ and maximize the distance to the irrelevant document group $\mathcal{R}^-$. Specifically, we employ negative log-likelihood as the loss function for positive references:

$$\mathcal{L}_G(q, \mathcal{R}^+, \mathcal{R}^-) \qquad (13)$$
$$= -\log \frac{\sum_{i=0}^{|\mathcal{R}^+|} e^{S(q, r_i^+)}}{\sum_{i=0}^{|\mathcal{R}^+|} e^{S(q, r_i^+)} + \sum_{i=0}^{|\mathcal{R}^-|} e^{S(q, r_i^-)}}.$$

**Pair-wise contrastive learning.** Most of the existing retrieval methods (Qin et al., 2022; Hossain et al., 2020) obtain the relevant documents by ranking the candidate documents based on the similarity to the query. They only minimize the negative log-likelihood of the positive documents, implicitly optimizing their model to binary classify the documents. However, the ranks of documents also provide a training signal to optimize the retrieval module. Specifically, we propose a pair-wise contrastive loss $\mathcal{L}_P$ which contrasts each positive reference $r_i^+$ in $R^+$ and negative reference $r_i^-$ in $R^-$, explicitly optimizing the model to rank $r_i^+$ before $r_i^-$. The $\mathcal{L}_P$ can be formulated as:

$$\mathcal{L}_P(q, \mathcal{R}^+, \mathcal{R}^-) \qquad (14)$$
$$= \sum_{i=0}^{|\mathcal{R}^+|} \sum_{j=0}^{|\mathcal{R}^-|} \max(0, S(q, r_j^-) - S(q, r_i^+) + \lambda),$$

where the $\lambda$ is the margin that we defined to explicitly constrain the gap between positive and negative references. Our contrastive loss is the combination of the $\mathcal{L}_G$ and $\mathcal{L}_P$:

$$\mathcal{L}_{CL} = \mu \mathcal{L}_G + (1 - \mu)\mathcal{L}_P, \qquad (15)$$

where $\mu$ is the weight of two losses.

### 4.4 Instruction-tuning for generation

Instruction tuning has been shown to improve the performance and generalization of LLMs in complex tasks for supervising the LLMs following concrete instructions (Chung et al., 2022; Wei et al., 2021). Since the definition of the RWG task is complex and different from summarizing text directly, we design an instruction-tuning method to help LLM understand the approach of generating related work. Specifically, we describe the definition of the RWG task via the demonstration $\mathcal{D}_{GEN}$. The $\mathcal{D}_{GEN}$ paired with abstract $X$ and reference $\mathcal{R}$ are concentrated to construct the instruction $\mathcal{I}_{GEN}$, which can be formulated as:

$$\mathcal{I}_{GEN} = \text{Demonstration} : \{\mathcal{D}_{GEN}\}$$
$$\text{Input} : \{\text{abstract} X\}$$
$$\text{Reference} : \{\mathcal{R}\}.$$

We then leverage the LLM to generate related work on the condition of the $\mathcal{I}_{GEN}$, prompting LLM to understand the definition of the RWG task and generate the related work section logically, which is supervised via the instruction-tuning loss:

$$\mathcal{L}_{GEN} = -\sum_{t=1}^{|Y|} \log P_\varphi(y_i^{(t)}|y_i^{(<t)}, \mathcal{I}_{GEN}). \qquad (16)$$

The $Y$ is the ground truth related work of the target paper and $\varphi$ is the parameters shared with Equation 1.

### 4.5 Multi-task learning

We jointly optimize UR³WG with two tasks, *i.e.,* query extension in § 4.2 and instruction-following generation in § 4.4. The final training objective is defined as:

$$\mathcal{J} = \alpha \mathcal{L}_{QE} + \eta \mathcal{L}_{GEN}, \qquad (17)$$

where the $\alpha$ and $\eta$ are the hyper-parameters, denoting the weights of the two losses, respectively.

Table 1: Results on two datasets. We abbreviate ROUGE as R. We underline the best results in each category of baseline methods. All ROUGE scores have a 95% confidence interval as reported by the official ROUGE script.

| Methods | TAS2 Dataset | | | | TAD Dataset | | | |
|---|---|---|---|---|---|---|---|---|
| | R-1 | R-2 | R-L | R-SU | R-1 | R-2 | R-L | R-SU |
| *Multi-document summarization methods* | | | | | | | | |
| NES (Wang et al., 2019) | 26.04 | 3.39 | 22.46 | 6.14 | 26.13 | 3.24 | 23.20 | 6.18 |
| MGSum (Jin et al., 2020) | 25.54 | 3.75 | 23.16 | 6.49 | 27.49 | 4.79 | 25.21 | 7.29 |
| LexRank (Erkan and Radev, 2004) | 25.74 | 2.81 | 22.43 | 6.03 | 25.70 | 2.86 | 22.68 | 5.90 |
| BertSumEXT (Liu and Lapata, 2019) | 25.85 | 2.90 | 22.66 | 6.21 | 25.95 | 2.91 | 23.05 | 6.25 |
| BertSumABS (Liu and Lapata, 2019) | 25.45 | 3.82 | 23.04 | 6.39 | 27.42 | 4.88 | 25.15 | 7.22 |
| EMS (Zhou et al., 2021) | 26.17 | 4.16 | 23.63 | 6.67 | 28.21 | 5.15 | 25.74 | 7.56 |
| *Large language models* | | | | | | | | |
| Llama-7B (Touvron et al., 2023) | 18.28 | 2.05 | 15.29 | 4.01 | 15.45 | 1.62 | 13.48 | 2.91 |
| Claude | 27.59 | 3.88 | 24.99 | 7.28 | 30.69 | 5.11 | 28.00 | 9.04 |
| Vicuna-7B | 28.40 | 4.01 | 25.39 | 7.01 | 31.09 | 4.91 | 27.62 | 8.9 |
| Davinci-text-003 | 22.80 | 2.49 | 20.26 | 5.08 | 24.59 | 3.15 | 21.97 | 5.89 |
| ChatGPT | 29.13 | 4.08 | 25.59 | 7.89 | 31.41 | 5.34 | 28.15 | 9.33 |
| *Related work generation methods* | | | | | | | | |
| RRG (Chen et al., 2021) | 26.79 | 4.43 | 24.46 | 6.85 | 28.94 | 5.59 | 26.46 | 7.92 |
| TAG (Chen et al., 2022a) | 28.04 | 4.75 | 25.33 | 7.69 | 30.48 | 6.16 | 27.79 | 8.89 |
| UR³WG | **31.59** | **5.86** | **26.13** | **9.62** | **32.68** | **7.74** | **28.87** | **9.54** |
| *Ablation study* | | | | | | | | |
| - w/o $\mathcal{L}_{QE}$ | 28.30 | 4.25 | 24.84 | 7.35 | 31.05 | 5.77 | 27.64 | 8.68 |
| - w/o $\mathcal{L}_{GEN}$ | 30.42 | 5.02 | 25.11 | 8.10 | 30.77 | 4.97 | 26.78 | 8.25 |

# 5 Experiment Setup

## 5.1 Datasets

We conduct experiments on two widely-applied datasets: TAS2 and TAD (Chen et al., 2022a). Specifically, the TAS2 consists of 117,700 scientific publications from several fields, whereas TAD consists of 218,255 scientific publications from the computer science field. For each example in datasets, it includes (1) the abstract of the article, (2) a paragraph of the related work, and (3) the reference paper cited in this paragraph (four references per paragraph on average).

## 5.2 Evaluation metrics

Following Chen et al. (2018); Zhou et al. (2021); Chen et al. (2022a), we mainly employ ROUGE (1,2,L,SU) for evaluation. Since only using the ROUGE metric to evaluate generation quality can be misleading (Shi et al., 2023), we also consider the other metrics, including the lexicon-based metric, e.g., BLEU (Papineni et al., 2002) and semantic-based metrics, e.g., BERTScore (Zhang et al., 2019) and BARTScore (Yuan et al., 2021). We evaluate the retriever with Recall@K (k=5, 10, 20) metrics. The statistical significance of differences observed between the performance of two runs is tested using a two-tailed paired t-test at $\alpha = 0.01$ and $p < 0.05$.

we also conduct the human evaluation. Three well-educated Master students are invited to judge 40 randomly sampled examples with a three-scale in the four aspects: Relevance (Rel.), Fluency (Flu.), Coherence (Cohe.), Informativeness (Info.). The details of these aspects can be found in Appendix A.5.

## 5.3 Baselines

We mainly compare our UR³WG with three types of baselines: multi-document summarization methods, related work generation methods, and large language models.

The *multi-document summarization* baselines include extractive methods, such as *LexRank* (Erkan and Radev, 2004), *BertSumExt* (Liu and Lapata, 2019) and *NES* (Wang et al., 2019) as well as abstractive methods like *BertSumAbs* (Liu and Lapata, 2019) and *EMS* (Zhou et al., 2021). These baselines take the references as input. The *related work generation methods* include *RRG* (Chen et al., 2021) and *TAG* (Chen et al., 2022a), which take the abstract and references as input. The *large language models* include Llama-7B (Touvron et al., 2023), Vicuna [1], Claude [2], Davinci-text-003, and ChatGPT [3], which take the instruction $\mathcal{I}_{GEN}$ input.

We also compare our retrieval method LER with strong baselines, including traditional term-based

[1] https://github.com/lm-sys/FastChat
[2] https://www.anthropic.com/
[3] https://openai.com/blog/chatgpt

Table 2: The recall@K (K=5,10,20) score for retrieval methods on two datasets.

| Methods | TAS2 Dataset | | | TAD Dataset | | |
|---|---|---|---|---|---|---|
| | Recall@5 | Recall@10 | Recall@20 | Recall@5 | Recall@10 | Recall@20 |
| BM25 (Robertson et al., 2009) | 6.32 | 9.79 | 13.82 | 2.62 | 4.62 | 7.10 |
| DPR (Karpukhin et al., 2020) | 13.23 | 18.30 | 27.89 | 6.28 | 9.65 | 16.9 |
| DLR (Lin and Lin, 2022) | 23.68 | 25.77 | 30.57 | 15.61 | 18.87 | 27.95 |
| SPLADE (Formal et al., 2021) | 24.21 | 30.51 | 37.31 | 15.70 | 20.80 | 26.29 |
| ColBERT (Khattab and Zaharia, 2020) | 15.56 | 21.14 | 27.89 | 23.34 | 24.12 | 25.11 |
| Condenser (Gao and Callan, 2021) | 20.84 | 28.94 | 38.48 | 11.22 | 16.61 | 23.94 |
| UniCOIL (Lin and Ma, 2021b) | 15.35 | 19.33 | 24.25 | 8.74 | 11.60 | 15.16 |
| Lexicon-enhanced retrieval (**LER**) | **37.35** | **41.56** | **44.17** | **31.20** | **40.20** | **49.47** |
| *Ablation study* | | | | | | |
| - w/o $\mathcal{L}_{QE}$ | 29.30 | 33.70 | 37.28 | 23.47 | 23.22 | 27.77 |
| - w/o $\mathcal{L}_{P}$ | 33.24 | 39.15 | 40.21 | 25.41 | 38.01 | 45.14 |
| - w/o $S_l$ | 34.10 | 40.23 | 41.03 | 29.16 | 35.26 | 44.87 |

methods like BM25 (Robertson et al., 2009), and dense retrievers, such as DPR (Karpukhin et al., 2020), ColBERT (Khattab and Zaharia, 2020), Condenser (Gao and Callan, 2021), as well as lexicon-aware retrievers, e.g., DLR (Lin and Lin, 2022), SPLADE (Formal et al., 2021) and UniCOIL (Lin and Ma, 2021b). Since the abstract is too long with many irrelevant words, we take concatenation of the keywords extracted from the abstract as the query. To ensure the fairness of comparison, all the baselines are finetuned with the same datasets as our proposed method. More details about the baselines are provided in Appendix A.3.

## 5.4 Implementation details

We initialize the parameters with Llama-7B (Touvron et al., 2023). We vary the weight of contrastive learning loss $\mu$ in $\{0.5, 0.6, 0.7, 0.8, 0.9\}$, and find that 0.7 achieves the best performance. Following the Sun et al. (2022), we tune the weight $\alpha$ to 0.5, $\eta$ to 0.5 at initialization, respectively, and linearly decrease to 0.3 and 0.2. We optimize the model using AdamW optimizer with parameters $\beta_1 = 0.98$, $\beta_2 = 0.99$, the learning rate of $2e^{-5}$, and the weight decay coefficient of 0.01. The training of our model can be done within 28 hours on the TAD dataset and 17 hours on the TAS2 dataset using four NVIDIA A100 PCIE GPUs.

## 6 Result and analysis

### 6.1 Experiment result

**Generation performance evaluation.** Table 1 shows the details of the results. Overall, UR³WG achieves the best performance compared to the other baselines. For example, UR³WG gets ROUGE-1=32.68, ROUGE-2=7.74 in TAD

datasets, with 8.4% and 25.6% relative improvement compared to existing state-of-the-art baseline. As shown in Table 4, we also select several strong baselines for more comprehensive evaluation. The proposed UR³WG outperforms the strong baselines in terms of the lexicon-level and semantic-level metrics, e.g., pushing BLEU-1 to 23.22 and BERTScore to 0.70 in TAS2 dataset. These results demonstrate that our UR³WG fits well in generating high-quality related work with the assistance of curated instruction.

**Retrieval performance evaluation.** As shown in Table 2, our retrieval module outperforms the strong retrieval baselines by a large margin. For example, LER reaches Recall@5=37.35 in the TAS2 dataset with 13.14 absolute improvement compared to the state-of-the-art baseline SPLADE, which illustrates that our proposed lexicon-enhanced dense retrieval is more suitable for reference retrieval scenarios. We also consider several variant models for more comprehensive comparison in Section 7.

### 6.2 Human evaluation

Table 6 shows the results of the human evaluation. We find that the UR³WG outperforms the best RWG baselines TAG in four aspects, e.g., pushing *Relevance* to 2.77 (0.47 absolute improvement) in TAD dataset. We also observe that UR³WG achieves comparable and better performance with ChatGPT (175B) with only 7B parameters, indicating the effectiveness of our method. The average Kappa statistics for four evaluation metrics are 0.72, 0.70, 0.71, and 0.73, illustrating agreement among the annotators. More results details are provided in Appendix A.5.

Table 3: The recall score for retrieval on two datasets. We take the input abstract, extended abstract (denoted as abstract*) and keywords extracted from input abstract as the query to search relevant documents, respectively.

| Method | Query | TAS2 | | | TAD | | |
|---|---|---|---|---|---|---|---|
| | | Recall@5 | Recall@10 | Recall@20 | Recall@5 | Recall@10 | Recall@20 |
| ConDenser | abstract | 7.13 | 11.49 | 17.48 | 3.99 | 6.81 | 11.04 |
| | abstract* | 29.27 | 32.56 | 36.04 | 19.6 | 21.89 | 24.49 |
| | keywords | 11.22 | 16.61 | 23.94 | 20.84 | 28.94 | 38.48 |
| ColBERT | abstract | 2.93 | 5.16 | 8.07 | 1.91 | 3.31 | 5.34 |
| | abstract* | 32.98 | 33.92 | 35.33 | 23.34 | 24.12 | 25.11 |
| | keywords | 12.47 | 17.14 | 22.63 | 10.55 | 14.70 | 19.26 |
| SPLADE | abstract | 6.08 | 9.08 | 13.10 | 4.06 | 6.48 | 9.59 |
| | abstract* | 36.38 | 41.92 | 47.20 | 30.42 | 36.14 | 41.36 |
| | keywords | 24.21 | 30.51 | 37.31 | 15.70 | 20.80 | 26.29 |
| UniCOIL | abstract | 2.13 | 4.62 | 6.57 | 1.76 | 2.67 | 4.13 |
| | abstract* | 19.18 | 21.33 | 24.15 | 12.72 | 14.26 | 16.16 |
| | keywords | 15.35 | 19.33 | 24.25 | 8.74 | 11.60 | 15.16 |
| *Approximation* | | | | | | | |
| LER (w. approx.) | abstract* | 35.59 | 40.50 | 43.23 | 29.14 | 40.11 | 48.32 |
| LER | abstract* | 37.35 | 41.56 | 44.17 | 31.20 | 40.20 | 49.47 |

Table 4: More evaluation results. B1: BLEU-1; B2: BLEU-2; BES: BERTScore, BAS: BARTScore.

| Methods | TAS2 Dataset | | | | TAD dataset | | | |
|---|---|---|---|---|---|---|---|---|
| | B1 | B2 | BES | BAS↑ | B1 | B2 | BES | BAS↑ |
| TAG | 17.23 | 7.50 | 0.660 | -4.61 | 18.82 | 7.61 | 0.653 | -4.76 |
| Claude | 15.30 | 5.37 | 0.650 | -5.38 | 17.41 | 6.51 | 0.660 | -5.17 |
| Vicuna | 14.17 | 4.34 | 0.604 | -4.85 | 17.61 | 6.18 | 0.651 | -5.20 |
| Davinci-text-003 | 14.92 | 4.19 | 0.589 | -5.50 | 16.25 | 4.91 | 0.601 | -5.31 |
| ChatGPT | 16.77 | 5.18 | 0.645 | -5.37 | 15.93 | 5.25 | 0.654 | -5.20 |
| UR³WG | 23.22 | 8.71 | 0.700 | -3.74 | 24.31 | 7.64 | 0.674 | -4.30 |
| *Ablation study* | | | | | | | | |
| - w/o $\mathcal{L}_{QE}$ | 22.47 | 8.09 | 0.664 | -4.34 | 23.72 | 7.01 | 0.656 | -4.97 |
| - w/o $\mathcal{L}_{GEN}$ | 22.47 | 8.64 | 0.650 | -4.02 | 22.98 | 7.24 | 0.632 | -5.01 |

## 6.3 Ablation study

To better understand the impact of different components of our UR³WG, we employ the following modifications to the architecture.

**–w/o $\mathcal{L}_P$.** We remove the loss $\mathcal{L}_P$ mentioned in Equation 15. We observed an average decrease in recall of $8 \sim 10$ on both datasets, *e.g.,*, the Recall@10 drops from 41.56 to 39.15 with 6.16% relative decrease, 40.20 to 38.01 with 5.76% relative decrease, which indicates the effect of our multi-granularity contrastive.

**–w/o $S_l$.** We remove the lexicon matching score $S_l(q, d)$ (in § 4.3.2). From the results shown in Table 2, the recall metrics have a significant decline, which indicates that the lexicon information is critical to retrieve relevant references in the RWG task.

**–w/o $\mathcal{L}_{QE}$.** We remove the loss $\mathcal{L}_{QE}$ for query extension. As shown in Table 2 and Table 1, both the generation and retrieval performance suffers from a significant decrease. This result indicates that the external knowledge injected via $\mathcal{L}_{QE}$ assists the UR³WG in generating more informative pseudo

references and final related work section.

**–w/o $\mathcal{L}_{GEN}$.** We replace the instruction-tuning loss $\mathcal{L}_{GEN}$ with a standard language modeling objective (Zhao et al., 2020c; Johansen and Juselius, 2010), where we take the concatenation of abstract and references as input, supervising the model predict the ground-truth related work. As shown in Table 1, the performance significantly declines, demonstrating that the instruction we design fits the LLMs well in the RWG task and improves the capability in complex generation scenarios.

## 6.4 Case study

We conduct several case studies and find that UR³WG is more effective at retrieving relevant references and generating more informative text than baselines. Details are provided in Appendix A.6.

## 7 Discussion

**Variant model for reference retrieval.** For each retrieval, we take the abstract, keywords of abstract, and extended abstract as the query to search relevant documents. Specifically, the keywords are extracted via KeyBert (Grootendorst, 2020), and extended abstract is generated via the proposed UR³WG. As shown in Table 3, for each retrieval method, taking the extended abstract as the query reaches the best performance, which indicates that our *query extension* method can be combined with different retrieval methods. And this result further illustrates that the extended abstract contains rich information about the target paper. We also observe a significant decline when we replace the query

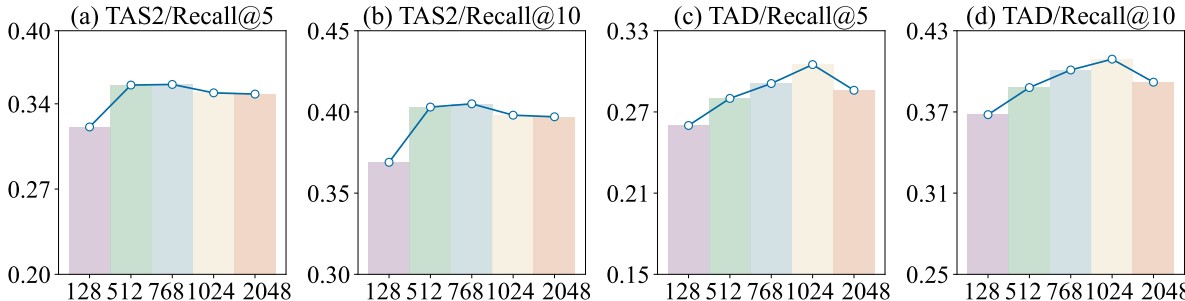

Figure 3: The retrieval performance of TAS2 dataset (**a** and **b**) and TAD dataset (**c** and **d**) based on different slice. For each figure, the horizontal axis indicates the **group size** (128, 512, 768, 1024, 2048) and the vertical axis indicates the corresponding **Recall@k** score (k=5, 10).

from abstract to keywords. The potential reason is that the abstract is too long with many irrelevant words, hindering the performance of retrieval.

**The impact of lexicon representation group size.** we conduct experiments to alternate the sparse lexicon representation into different dimensions. As shown in Figure 3, we find that the recall keeps increasing and peaks at 1024 (TAD datasets) and 768 (TAS2 datasets), indicating that more representative lexicon information can be used to match relevant documents in the retrieval stage. However, when the dimension increases to 2048, the recall decreases, suggesting that some low-featured lexicon information is retained.

**Qualitative analysis about the approximation** We explore the impact by comparing the performance before and after approximation, and results are are shown in Table 3. Comparing with the vanilla **LER**, we observe that **LER (w. approx.)** an average decrease of about 3.13% and 3.68% in TAS2 and TAD datasets, respectively. This result illustrates that the approximation can improve retrieval efficiency at a slight performance cost. More details can be found in Appendix A.2.

## 8 Conclusion

In this paper, we propose a UR$^3$WG, which fuses the reference retrieval and related work generation into a unified framework based on large language models (LLMs). Concretely, UR$^3$WG first extend the input abstract and generate the query used for next retrieval stage. Then a lexicon-enhanced dense retrieval (LER) is proposed to retrieve the relevant document at the semantic and lexicon level. The LER is optimized via multi-granularity contrastive learning, including group- and pair-wise contrasts.

Since RWG is different from simple text summarization tasks, we employ an instruction-tuning method, enabling the LLM to understand the definition of RWG task and generate a high-quality related work section. Experiments on two datasets demonstrate that our model establishes a new state-of-the-art with automatically retrieved references.

## Limitations

The main limitation of this paper is the requirement of massive computation resource. We will explore how to employ our method in low-resource scenarios. Another limitation is that we only consider the single-lingual retrieval scenario. In the future, we plan to extend our model to multi-lingual applications, which is common in real-world.

## Ethics Statement

The paper proposes UR$^3$WG, a unified reference retrieval and related work generation framework. Given the abstract as input, the UR$^3$WG first searches relevant papers as references and generate related work. Moreover, reference papers should be cited appropriately.

## Acknowledgements

This work was supported by the National Key R&D Program of China with grant No. 2020YFB1406704, the Natural Science Foundation of China ( T2293773, 62272274, 61972234, 62072279, 62102234, 62202271), the Natural Science Foundation of Shandong Province (ZR2022QF004), the Key Scientific and Technological Innovation Program of Shandong Province (2019JZZY010129), the Fundamental Research Funds of Shandong University.

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

## A  Appendix

### A.1  Retrieving on large-scale corpus

Since the scientific paper corpus is usually large-scale in the real-world scenario, our UR³WG should support a billion-scale similarity search more efficiently. Following the Lewis et al. (2020), we first encode all candidate documents into vector representations using the encoder $Encoder_d$, and utilize Faiss to store and conduct the real-time vector search. Faiss[4] is a library for efficient similarity search and clustering of dense vectors. Faiss employs the MIPS (Maximum Inner Product Search) to find the document vectors with the highest inner product with the query vector (Mussmann and Ermon, 2016).

Since the online interaction mentioned in Equation 11 is time-intensive, we can approximate it by creating an index of group-wise maximum and minimum values of $\theta$:

$$\tau_d = \{\arg\max(\theta_d^{i\sim i+k})|i = nk, n \le \lfloor \frac{|\mathcal{V}|}{k} \rfloor\},$$

$$\widetilde{\theta}_{d,max} = \{\max(\theta_d^{i\sim i+k})|i = nk, n \le \lfloor \frac{|\mathcal{V}|}{k} \rfloor\},$$

$$\widetilde{\theta}_{d,min} = \{\min(\theta_d^{i\sim i+k})|i = nk, n \le \lfloor \frac{|\mathcal{V}|}{k} \rfloor\},$$

where the $\lfloor \cdot \rfloor$ indicates the lower rounding.

The essence of our approximation method is that the bag-of-words representation is highly sparse, so we can assume that only one individual element in each group is significantly larger than the other elements, while the other elements are only slightly different. Therefore, the Equation 11 can be further simplified as:

$$S_l = \sum_{i=0}^{|\tau_q|} \sigma_i \cdot \widetilde{\theta}_q^i \cdot \widetilde{\theta}_{d,max}^i + (1 - \sigma_i) \cdot \widetilde{\theta}_q^i \cdot \widetilde{\theta}_{d,min}^i$$

where $\sigma_i$ is a label defined as:

$$\sigma_i = \begin{cases} 0, & \tau_q^i \ne \tau_d^i \\ 1, & \tau_q^i = \tau_d^i \end{cases} . \quad (18)$$

For $i$-th group of $\theta_d$ during the matching, if the $\tau_d^i$ equals to $\tau_q^i$, there is no bias. If the $\tau_d^i$ not equals to $\tau_q^i$, the value $\tau_d^i$-th element in $\theta_d$ can be alternate by the minimum value of this group based on our assumption. With this approximation, we can use Faiss to perform semantic matching on the large-scale corpus.

---

[4]https://github.com/facebookresearch/faiss

### A.2  Qualitative analysis about the approximation

To support the retrieval on the large-scale corpus, we approximate the interaction to adapt the vector retrieval method in Faiss (in Eq A.1), which unavoidably introduces the loss of information. We explore the impact by comparing the performance before and after approximation, and the results are shown in Table 3 as **LER (w. approx.)**. Comparing with the vanilla **LER** , we can observe that the performance of the algorithm has decreased, with an average decrease of about 3.13% and 3.68% in TAS2 and TAD datasets, respectively. This result illustrates that the approximation can improve retrieval efficiency at a slight performance cost. With this approximation, we can use Faiss to perform semantic matching on the large-scale corpus. Otherwise, it will be difficult for us to retrieve on a large-scale corpus with acceptable latency.

### A.3  Details for baselines

We mainly compare our UR³WG with three types of baselines: multi-document summarization methods, related work generation methods, and large language models.

The *multi-document summarization* baselines include: *LEAD*, which concatenates the first sentence of each reference; *LexRank* (Erkan and Radev, 2004), which extract summarization based on the graph representation of sentences; *NES* (Wang et al., 2019), a extractive method which measures the relevance via bibliography graph; *MGSum* (Jin et al., 2020), an abstractive method based on multi-granularity interaction network; *BertSum* (Liu and Lapata, 2019), a summarization system built on a pre-trained BERT (Devlin et al., 2019); *EMS* (Zhou et al., 2021), which augments the encoder-decoder framework with a heterogeneous graph.

The *related work generation methods* include: *RRG* (Chen et al., 2021) , which enhanced by a iteratively refined relation-aware graph between references; *TAG* (Chen et al., 2022a), a transformer-based model with a target-aware graph encoder.

The *large language models* (LLMs) include flan-t5 (Chung et al., 2022), Llama-7B (Touvron et al., 2023), ChatGLM (Du et al., 2022), Vicuna [5], claude [6], Davinci-text-003, and ChatGPT [7], where the instruction $\mathcal{I}_{GEN}$ is taken as input.

---

[5]https://github.com/lm-sys/FastChat
[6]https://www.anthropic.com/
[7]https://openai.com/blog/chatgpt

Table 5: Results on the TAS2 and TAD dataset. We abbreviate ROUGE as R We underline the best results in each category of baseline methods. w/ T denotes the method which uses both references and abstract as input. All ROUGE scores have a 95% confidence interval as reported by the official R script.

| Methods | TAS2 Dataset | | | | TAD Dataset | | | |
|---|---|---|---|---|---|---|---|---|
| | R-1 | R-2 | R-L | R-SU | R-1 | R-2 | R-L | R-SU |
| *Summarization methods* | | | | | | | | |
| LEAD | 19.63 | 1.75 | 16.88 | 3.73 | 22.74 | 2.32 | 20.15 | 4.87 |
| NES **w/ T** (Wang et al., 2019) | 26.04 | 3.39 | 22.46 | 6.14 | 26.13 | 3.24 | 23.20 | 6.18 |
| MGSum (Jin et al., 2020) | 25.54 | 3.75 | 23.16 | 6.49 | 27.49 | 4.79 | 25.21 | 7.29 |
| LexRank (Erkan and Radev, 2004) | 25.74 | 2.81 | 22.43 | 6.03 | 25.70 | 2.86 | 22.68 | 5.90 |
| LexRank **w/ T** (Erkan and Radev, 2004) | 27.04 | 3.18 | 23.48 | 6.55 | 27.29 | 3.50 | 24.06 | 6.61 |
| BertSumEXT (Liu and Lapata, 2019) | 25.85 | 2.90 | 22.66 | 6.21 | 25.95 | 2.91 | 23.05 | 6.25 |
| BertSumEXT **w/ T** (Liu and Lapata, 2019) | 27.43 | 3.56 | 24.01 | 6.97 | 27.60 | 3.64 | 24.51 | 6.98 |
| BertSumABS (Liu and Lapata, 2019) | 25.45 | 3.82 | 23.04 | 6.39 | 27.42 | 4.88 | 25.15 | 7.22 |
| BertSumABS **w/ T** (Liu and Lapata, 2019) | 26.35 | 3.23 | 23.73 | 6.65 | 28.48 | 3.93 | 25.91 | 7.63 |
| EMS (Zhou et al., 2021) | 26.17 | 4.16 | 23.63 | 6.67 | 28.21 | 5.15 | 25.74 | 7.56 |
| EMS **w/ T** (Zhou et al., 2021) | 26.50 | 4.22 | 23.90 | 6.84 | 28.74 | 5.36 | 26.37 | 7.89 |
| *Large language models* | | | | | | | | |
| Llama-7B (Touvron et al., 2023) | 18.28 | 2.05 | 15.29 | 4.01 | 15.45 | 1.62 | 13.48 | 2.91 |
| Claude | 27.59 | 3.88 | 24.99 | 7.28 | 30.69 | 5.11 | 28.00 | 9.04 |
| Vicuna-7B | 28.40 | 4.01 | 25.39 | 7.01 | 31.09 | 4.91 | 27.62 | 8.9 |
| Davinci-text-003 | 22.80 | 2.49 | 20.26 | 5.08 | 24.59 | 3.15 | 21.97 | 5.89 |
| ChatGPT | 29.13 | 4.08 | 25.59 | 7.89 | 31.41 | 5.34 | 28.15 | 9.33 |
| *Related work generation methods* | | | | | | | | |
| RRG (Chen et al., 2021) | 26.79 | 4.43 | 24.46 | 6.85 | 28.94 | 5.59 | 26.46 | 7.92 |
| TAG (Chen et al., 2022a) | 28.04 | 4.75 | 25.33 | 7.69 | 30.48 | 6.16 | 27.79 | 8.89 |
| UR³WG | **31.59** | **5.86** | **26.13** | **9.62** | **32.68** | **7.74** | **28.87** | **9.54** |

We compare our lexicon-enhance retrieval method with the following baselines: BM25 (Robertson et al., 2009), a classical sparse retrieval; DPR (Karpukhin et al., 2020), a dense retrieval method with a dual-encoder framework; SPLADE (Formal et al., 2021), a passage ranker based on sparsity regularization and a log-saturation effect; ColBERT (Khattab and Zaharia, 2020), which searches relevant passages via contextualized late interaction. UniCOIL (Lin and Ma, 2021b), an extension of classical sparse retrieval COIL (Gao et al., 2021). Since the abstract is too long with many irrelevant words, we take concatenation of the keywords extracted from the abstract as the query.

## A.4 Variant model

Since the existing summarization baselines only use the references as input, the abstract of the target paper also contains much useful information. To make a fair comparison, we develop a variant model for each multi-document summarization method which uses the concatenation of the abstract of the target paper and references as input. We add the notion (**w/ T**) to each original *summarization* method to denote the variant model. We also implement a baseline, denoted as *LEAD*, which concatenates the first sentence of each ref-

erence. As shown in Table 5, the ROUGE metrics increase when the abstract is concatenated to the reference as input, e.g., an average 4% relative improvement in ROUGE-1 metrics. However, it still lags behind the proposed UR³WG.

## A.5 Human evaluation

### A.5.1 Details for human evaluation

We conduct human evaluation, where three well-educate Master students are invited to judge 40 randomly sampled examples. Specifically, we show each annotator the abstract, related work, and corresponding references. Each annotator is asked to rate the related work with a three-scale in the following metrics: (1) Relevance (Rel.): the relevance between the raw abstract and the related work. (2) Fluency (Flu.): whether the related work is fluent with no grammatical errors. (3) Coherence (Cohe.): coherence refers to how well the sentences are connected and how they flow together to form a coherent and understandable text. (4) Informativeness (Info.): Whether related work covers the key information mentioned in the article or can provide more extensions. Table 6 shows the details of the human evaluation results.

Table 6: Human evaluation results on two datasets.

| Methods | TAD dataset | | | | TAS2 dataset | | | |
|---|---|---|---|---|---|---|---|---|
| | Rel. | Flu. | Cohe. | Info. | Rel. | Flu. | Cohe. | Info. |
| TAG (Chen et al., 2022a) | 2.30 | 2.50 | 2.43 | 2.25 | 2.54 | 2.40 | 2.55 | 2.35 |
| Claude | 2.15 | 2.70 | 2.45 | 2.33 | 2.35 | 2.77 | 2.35 | 2.39 |
| Vicuna | 2.25 | 2.75 | 2.32 | 2.35 | 2.34 | 2.77 | 2.44 | 2.47 |
| Davinci-text-003 | 1.90 | 2.51 | 1.97 | 1.97 | 2.05 | 2.60 | 2.01 | 2.05 |
| ChatGPT | 2.65 | **2.86** | 2.40 | **2.65** | 2.54 | **2.83** | 2.44 | 2.65 |
| **UR³WG** | **2.77** | 2.79 | **2.45** | 2.60 | **2.57** | 2.70 | **2.54** | **2.70** |
| Human | 2.84 | 2.85 | 2.78 | 2.89 | 2.79 | 2.74 | 2.95 | 2.75 |

Table 7: An example of generated related work by our model and baselines. The same color denotes the same key information.

---

**Abstract of target paper (truncated):**
we explore a simple method for **predicting the genre of artists** based on **co-occurrence analysis**, i.e., analyzing co-occurrences of artist and genre names on music related web pages. We investigate four different query schemes and two probabilistic approaches for predicting the genre of artists.

**Ground Truth Reference Papers (the same as our retrieved reference):**
[1] **Detecting artists** that can be considered as prototypes for particular genres or styles of music is an interesting task.
[2] We use the relation between the prototypicality ranking and a ranking obtained by **page counts** of search to google that contain artist and genre.

---

**Ground truth related work** : **Co-occurrence analysis** has also been applied to the **detecting prototypical artists** task for a given genre[1]. In [2], they used a technique based on a page rank citation ranking on **page count** estimates to derive the prototypicality of each of artist for a given genre.

---

**Baseline TAG output**: **Co-occurrences** of artist names on web pages have been used for artist co-occurrence analysis in [1]. In [2], the authors use the co-occurrence analysis of **artist and genre names** to determine artist similarity. They use cosine similarity to find co-occurrences.

---

**Baseline ChatGPT output**: This survey focuses on a web-based **Co-occurrences** technique for genre prediction of **music artists**. By evaluating the effectiveness of this approach and considering the integration of objective rating methods like **Pagerank**, it aims to contribute to the development of accurate and reliable **genre** prediction models.

---

**UR³WG output**: [1] use **co-occurrences of artist and genre names** on music related web pages to **predict the genre of a given artist**. [2] propose a method for predicting the genres of an artist which relies on the **page counts** provided by google to estimate the relatedness of an arbitrary artist to each of a set of genres.

---

## A.6 Case study

Table 7 presents an example of the generated related work by UR³WG and the start-of-the-art baseline TAG. And we highlight the same key information with the same color. We first observe that UR³WG retrieves the ground-truth references correctly, even though the keyword "page rank" is not mentioned in the abstract, which verifies the efficiency of our retriever. Moreover, the related work generated by UR³WG captures more highly related key information than the TAG baseline.