# OpenReview forum: "Towards a Unified Framework for Reference Retrieval and Related Work Generation"
_EMNLP/2023/Conference — EMNLP 2023 Findings_

### Official Review · Reviewer_FLmC · 2023-07-31

**Soundness:** 3

**Excitement:**

4: Strong: This paper deepens the understanding of some phenomenon or lowers the barriers to an existing research direction.

**Paper Topic And Main Contributions:**

This paper proposes a framework unifying reference retrieval and related work generation based on LLMs. The proposed method reformulate the query and retrieval the documents as references for better related work generation. Specifically, the lexicon-enhanced retriever significantly helps the retrieved documents, thereby delivering better summarization.

**Reasons To Accept:**

1. Smart unified framework to combine related work generation and retrieval.
2. This paper is well-written and easy to follow.
3. The experimental results are strong.

**Reasons To Reject:**

1. Missing several important lexicon-aware baselines in Table 3 like SPLADE v2, SPLADE ++, and LED. It's a bit weird to me that there’s no methods after 2021 in Table 3.

**Reproducibility:**

4: Could mostly reproduce the results, but there may be some variation because of sample variance or minor variations in their interpretation of the protocol or method.

**Reviewer Confidence:**

4: Quite sure. I tried to check the important points carefully. It's unlikely, though conceivable, that I missed something that should affect my ratings.

---

> ### Author Rebuttal · Authors · 2023-08-29
>
> Thank you for the valuable comments that help us improve the work. Below we address the concerns mentioned in the review:
>
> We provide the comparison with three recent baselines, such as SPLADE v2[1] and SPLADE ++[2]. And the Table 8 summarizes the results. We find that our method outperforms the baslines in two datasets. For example, the LER achieves Recall@5=37.35 , with 12.87 absolute improvement compared to SPLADE++ in the TAD dataset. These results further illustrate the effectiveness of our retrieval method.
>
> We will add these baselines to our main experiment section in the final version. And it would be very kind of you to raise the current score if you find our responses useful and helpful.
>
> &nbsp;
>
> Table 8: The recall score for retrieval on two datasets. We compare with more recent baselines to further examine the effectiveness of our method.
> |           |          | TAS2     | Dataset   |          | TAD | Dataset   |
> |------|:---------:|:--------:|:---------:|:---------:|:--------:|:---------:|
> |           | Recall@5 | Recall@10 | Recall@20 | Recall@5 | Recall@10 | Recall@20 |
> | DLR[3]      |   23.68 |   25.77     |     30.57     |  15.61  |  18.87  |      27.95   |
> | SPLADE v2[1]|    27.67 |  33.95     |     40.68     |  14.88   |   19.41   |     24.50    |
> | SPLADE ++[2]|     24.48  |   29.64  |    34.87   |  16.10  |   20.77    |    41.10  |
> | LER         |    37.35  |   41.56     |     44.17     |  31.20   |   40.20    |      49.47     |
>
> [1] SPLADE v2: Sparse Lexical and Expansion Model for Information Retrieval
>
> [2] From Distillation to Hard Negative Sampling: Making Sparse Neural IR Models More Effective
>
> [3] A Dense Representation Framework for Lexical and Semantic Matching

---

### Official Review · Reviewer_5RZj · 2023-08-02

**Typos Grammar Style And Presentation Improvements:** N/A
**Soundness:** 3

**Excitement:**

2: Mediocre: This paper makes marginal contributions (vs non-contemporaneous work), so I would rather not see it in the conference.

**Missing References:**

A Dense Representation Framework for Lexical and Semantic matching

**Paper Topic And Main Contributions:**

This propose a Unified Reference Retrieval and Related Work Generation Model (URWG), which combines reference retrieval and related work generation processes in a unified framework based on the large language model for the related work generation task. Experiments on two wide-applied datasets demonstrate that URWG outperforms the baselines in both generation and retrieval metrics.

**Questions For The Authors:**

See Reasons To Reject

**Reasons To Accept:**

(1) Propose the first unified reference retrieval and related work generation model.
(2) Experiments demonstrate URWG outperforms the baselines in both generation and retrieval metrics.

**Reasons To Reject:**

(1) Similar technologies without proper comparison. The authors claim that they propose Lexicon-Enhance Dense Retrieval (LER), however, the technologies in LER is very similar to DLRs [1]. I don't think these two can be considered as contemporaneous work, and hope that the authors make proper clarifications and citations of the duplicated technical points, instead of declaring these as their contributions as well.

(2) Limited technical contribution.



[1] A Dense Representation Framework for Lexical and Semantic Matching

**Reproducibility:**

4: Could mostly reproduce the results, but there may be some variation because of sample variance or minor variations in their interpretation of the protocol or method.

**Reviewer Confidence:**

4: Quite sure. I tried to check the important points carefully. It's unlikely, though conceivable, that I missed something that should affect my ratings.

---

> ### Author Rebuttal · Authors · 2023-08-29
>
> Thank you for the valuable comments that help us improve the work. Below we address the concerns mentioned in the review:
>
>
> **Question 1: Comparison with previous retrieval method.**
>
> The DLRs[1] is a classical lexicon-enhance retrieval method that shares a similar motivation with approaches such as LexMAE[2], SPLADE[3], and ColBERT[4]. Inspired by these methods, we propose a novel method LER to measure the relevance between query and document on both semantic and lexical levels.
>
> To tackle the specific challenges in the RWG task, there are three main contributions in our proposed retrieval method:
> 1. Considering the user-provided abstract is vague without a clear intention for reference, we cannot directly take the abstract as the retrieval query. We leverage the knowledge of the LLM to re-generate an extended abstract that contains more related topics inherently correlated with the target paper.
> 2. Scientific papers entail multiple diverse references, in contrast to the existing passage retrieval task that assumes a single relevant document per query. Therefore, we propose a multi-granularity contrastive learning method to train our retriever to tackle this problem.
> 3. Due to the size of the scientific paper corpus is usually large and it is time-intensive to conduct the interaction (Equation 12) in real-world applications. We design an approximate algorithm for semantic matching.
>
> As you suggested, we further compare with the DLRs in Table 8 for a more comprehensive analysis. We also include two more recent retrieval methods: SPLADE v2 and SPLADE++. From Table 8, we can find that our proposed LER outperforms these baselines, such as achieving 13.81 absolute improvement  compared with SPLADE v2 on average. Since the DLRs is also a typical lexicon-enhance retrieval method, we will definitely add this citation in our final version.
>
> &nbsp;
>
> Table 8: The recall score for retrieval on two datasets. We compare with more recent baselines to further examine the effectiveness of our method.
> |           |          | TAS2     | Dataset   |          | TAD | Dataset   |
> |------|:---------:|:--------:|:---------:|:---------:|:--------:|:---------:|
> |           | Recall@5 | Recall@10 | Recall@20 | Recall@5 | Recall@10 | Recall@20 |
> | DLR[1]      |   23.68 |   25.77     |     30.57     |  15.61  |  18.87  |      27.95   |
> | SPLADE v2[5]|    27.67 |  33.95     |     40.68     |  14.88   |   19.41   |     24.50    |
> | SPLADE ++[6]|     24.48  |   29.64  |    34.87   |  16.10  |   20.77    |    41.10  |
> | LER         |    37.35  |   41.56     |     44.17     |  31.20   |   40.20    |      49.47     |
>
> &nbsp;
>
> **Question 2: Limited technical contribution.**
>
> Our work aims to build a unified framework for relation work generation task, which consists of two main parts: reference retrieval and related work generation.
>
> Since the reference retrieval in RWG task has some specific challenges, the existing dense retrieval methods cannot suit the RWG task well. As introduced in the response to Question #1, we propose the LER model to retrieve more relevant references in this scenario.
>
> In the related work generation module, according to the definition of related work generation task[7,8], the related work section should comprehensively introduce the correlation between the references, instead of simply summarizing the primary points for multiple documents. We propose an instruction-tuning based generation method to generate a high-quality related work section, which leverages the instruction-following capability of LLM.
>
> We will add these details for a clearer explanation and highlight the contribution of our work in the camera-ready version. It would be very kind of you to raise the current score if you find our responses useful and helpful.
>
> &nbsp;
>
> [1] A Dense Representation Framework for Lexical and Semantic Matching
>
> [2] LexMAE: Lexicon-Bottlenecked Pretraining for Large-Scale Retrieval
>
> [3] SPLADE: Sparse lexical and expansion model for first stage ranking
>
> [4] ColBERT: Efficient and Effective Passage Search via Contextualized Late Interaction over BERT
>
> [5] SPLADE v2: Sparse Lexical and Expansion Model for Information Retrieval
>
> [6] From Distillation to Hard Negative Sampling: Making Sparse Neural IR Models More Effective
>
> [7] Towards Automated Related Work Summarization
>
> [8] Target-aware Abstractive Related Work Generation with Contrastive Learning

---

### Official Review · Reviewer_Vyrb · 2023-08-06

**Soundness:** 3

**Excitement:**

3: Ambivalent: It has merits (e.g., it reports state-of-the-art results, the idea is nice), but there are key weaknesses (e.g., it describes incremental work), and it can significantly benefit from another round of revision. However, I won't object to accepting it if my co-reviewers champion it.

**Paper Topic And Main Contributions:**

The paper proposes a unified, joint learning framework UR3WG for Reference Retrieval and Related Work Generation.  The framework takes three steps. Given a user query (e.g. abstract of a paper), UR3WG first learns to generate an extended query, retrieve related work, and generate a summary for all the related work.

The paper presents SOTA results on TAS2 and TAD datasets.

**Questions For The Authors:**

See Reasons to reject above -- Happy to hear what the authors think, and will adjust my ratings accordingly.

**Reasons To Accept:**

The proposed framework presents state of the art results on two of the related work generation datasets.
Using the same backbone model for query extension and related work generation in a joint learning setting is an interesting idea, and has some intellectual merit.
The paper presents solid evaluation results and ablation studies for us to understand the performance of the model.

**Reasons To Reject:**

Though the proposed framework presents state-of-the-art results, it's a bit hard to tell from the evaluation in its current form if the improvements can be attributed to the conceptual idea, or LLM itself. Specifically, IMHO here are some comments and suggestions on where the authors can potentially improve their evaluations.

- For reference retrieval, the LER system seems to be the only one that's finetuned on scientific papers specifically, all the other retrieval baselines are evaluated in zero-shot settings. So it's hard to tell if the LER design specifically is contributing to the improvements or not.
- Given the relatively small improvements in terms of ROUGE over ChatGPT (especially considering the ++ amount of supervision that the proposed system used compared to LLM baselines), it might be worth doing more analyses and justify why we should use your system instead of ChatGPT for related work generation in general. One way to strengthen the evaluation could be doing a cross-domain generalization study on your system, e.g. train on one dataset and evaluate on the other, so that we can see if your system is able generalize or not.
- Along the previous point, the human evaluation presented in Table 6  in Appendix seems to favor vanilla ChatGPT over your proposed system even for in-domain train+eval setting.  This is fine, but it's worth dedicate more discussion + analyses on why it could be in the main body of the paper?

**Reproducibility:**

3: Could reproduce the results with some difficulty. The settings of parameters are underspecified or subjectively determined; the training/evaluation data are not widely available.

**Reviewer Confidence:**

4: Quite sure. I tried to check the important points carefully. It's unlikely, though conceivable, that I missed something that should affect my ratings.

**Typos Grammar Style And Presentation Improvements:**

Overall I feel like the paper could use more space for results and error case analysis in the main body of the paper, and truncate/trim the description on the methods.

---

> ### Author Rebuttal · Authors · 2023-08-29
>
> Thank you for the valuable comments that help us improve the work. Below we address the concerns mentioned in the review.
>
> **Question 1: Fairness of comparison to baselines.**
>
> To ensure the fairness of comparison, all the retrieval baselines in Table 3 are finetuned with the same datasets as our proposed LER.
> We will add these details in the camera-ready version.
>
> **Question 2.1: More analyses and justify why we should use your system instead of ChatGPT for related work generation in general.**
>
> Despite the ChatGPT achieving remarkable performance in the related work generation task, it also encounters the following limitations:
> 1. For data security reasons[1,2], not all applications can transmit user data, such as reference papers and unpublished content, to LLM service vendors. It restricts the use of proprietary LLMs in such applications.
> 2. The proprietary large language model, e.g., ChatGPT, is cost-intensive during inference for its huge parameters, leading to a tremendous carbon footprint. According to the [3] and [4], the ChatGPT exhibits a carbon footprint that is ten times greater than Llama-7B. In contrast, our system achieves higher performance while utilizing only 4% parameters of ChatGPT. Given that our method is model-agnostic, it is intuitive that the performance of our system will improve along with the increase of the parameter scale. Considering the environmental concern posed by carbon emissions[3-6], we only use Llama-7B for validation within our research.
>
> **Question 2.2: The cross-domain evaluation.**
>
> We further conduct the cross-domain evaluation, where we train our model with one dataset and evaluate it in another. As shown in Table 8, our proposed method exhibits potential generalizability. For example, the *UR3WG w/cross* outperforms Claude with a 9.2%~11.5% average relative improvement on the two datasets. Moreover, our method achieves comparable and even better performance compared with ChatGPT. These results illustrate the capacity of our system to generalize across different related work generation datasets effectively.
>
> Table 8: Results on two datasets, where the R is the  abbreviation of ROUGE. We conduct the cross evaluation, where we train the model on one dataset and evaluate the other dataset (UR3WG w/cross).
> |                  |        | TAS2 | Dataset |      |       |  TAD | Dataset |      |
> |----------------|:------:|:----:|:-------:|:----:|:-----:|:----:|:-------:|:----:|
> |                  |   R-1  |  R-2 |   R-L   | R-SU |  R-1  |  R-2 |   R-L   | R-SU |
> |      Claude      |  27.59 | 3.88 |  24.99  | 7.28 | 30.69 | 5.11 |  28.00  | 9.04 |
> |     Vicuna-7B    | 28.40  | 4.01 | 25.39   | 7.01 | 31.09 | 4.91 | 27.62   | 8.9  |
> | Davinci-text-003 | 22.80  | 2.49 | 20.26   | 5.08 | 24.59 | 3.15 | 21.97   | 5.89 |
> | ChatGPT          | 29.13  | 4.08 | 25.59   | 7.89 | 31.41 | 5.34 | 28.15   | 9.33 |
> | UR3WG            | 31.59  | 5.86 | 26.13   | 9.62 | 32.68 | 7.74 | 28.87   | 9.54 |
> | UR3WG w/cross    | 30.01  | 4.77 | 25.50   | 8.34 | 31.75 | 6.74 | 27.79   | 9.27 |
>
>
> **Question 3: More discussion for the human evaluation.**
>
> To further examine the effectiveness of our method, we employ a harder baseline which feeds the ground truth references into ChatGPT to get a more competitive baseline, while our method only takes the model retrieved references as input. Notably, employing ground truth references yields better performance than utilizing retrieval references. Due to space constraints, we only present the superior result (where ground truth references are used as input) in the paper. For a more comprehensive comparison, the results with model retrieval reference as input are shown in Table 9 and Table 10. We can observe that our method outperforms baselines with the same setting in terms of automatic and human metrics.
> We will add these experimental details in the final version in the main body of the paper.
>
> &nbsp;
>
> Table 9: We feed ChatGPT with two types of reference, i.e., retrieval references (denoted as ChatGPT w/retrieval) and ground truth references (denoted as ChatGPT w/oracle).
> |                     |       | TAS2 | Dataset |      |       | TAD  | Dataset |      |
> |----------------|:------:|:----:|:-------:|:----:|:-----:|:----:|:-------:|:----:|
> |                     | R-1   | R-2  | R-L     | R-SU | R-1   | R-2  | R-L     | R-SU |
> | ChatGPT w/retrieval | 26.48 | 3.42 | 23.90   | 6.92 | 28.45 | 4.51 | 25.60   | 7.99 |
> | ChatGPT w/oracle    | 29.13 | 4.08 | 25.59   | 7.89 | 31.41 | 5.34 | 28.15   | 9.33 |
> | UR3WG               | 31.59 | 5.86 | 26.13   | 9.62 | 32.65 | 7.74 | 28.87   | 9.54 |
>
> &nbsp;
>
> Table 10: Human evaluation results on two datasets.
> |                     |      | TAS2 | Dataset |       |      | TAD  | Dataset |       |
> |----------------|:------:|:----:|:-------:|:----:|:-----:|:----:|:-------:|:----:|
> |                     | Rel. | Flu. | Cohe.   | Info. | Rel. | Flu. | Cohe.   | Info. |
> | ChatGPT w/retrieval | 2.59 | 2.70 | 2.24    | 2.50  | 2.47 | 2.55 | 2.14    | 2.52  |
> | ChatGPT w/oracle    | 2.65 | 2.86 | 2.47    | 2.65  | 2.54 | 2.83 | 2.44    | 2.70  |
> | UR3WG               | 2.64 | 2.75 | 2.35    | 2.60  | 2.57 | 2.70 | 2.30    | 2.60  |
> | Human               | 2.84 | 2.85 | 2.78    | 2.89  | 2.79 | 2.74 | 2.95    | 2.75  |
>
> &nbsp;
>
> We will also add these experimental results and discussion in the final version of the paper. It would be very kind of you to raise the current score if you find our responses useful and helpful.
>
> &nbsp;
>
> [1] The False Promise of Imitating Proprietary LLMs
>
> [2] WebCPM: Interactive Web Search for Chinese Long-form Question Answering
>
> [3] Reducing the Carbon Impact of Generative AI Inference (today and in 2035)
>
> [4] ChatGPT Needs SPADE (Sustainability, PrivAcy, Digital divide, and Ethics) Evaluation: A Review
>
> [5] Measuring the environmental impacts of artificial intelligence compute and applications
>
> [6] Artificial intelligence and the climate emergency: Opportunities, challenges, and recommendations

---

### Meta-Review · Area_Chair_Hpf5 · 2023-09-25

**Recommendation:** 3

**Metareview:**

The proposed method reformulate the query and retrieval the documents as references for better related work generation. The method is smart and useful. I personally respect the three reviewers' opinion, and suggest the paper accepted as Findings in the conference.

---

### Decision · Program_Chairs · 2023-10-07

**Decision:**

Accept-Findings

**Comment:**

The proposed method reformulate the query and retrieval the documents as references for better related work generation. The method is smart and useful. I personally respect the three reviewers' opinion, and suggest the paper accepted as Findings in the conference.